# Layer Grafted Pre-training: Bridging Contrastive Learning and Masked Image Modeling for Label-Efficient Representations

**Ziyu Jiang**[†,‡,*]**, Yinpeng Chen**[‡]**, Mengchen Liu**[‡]**, Dongdong Chen**[‡]**, Xiyang Dai**[‡]**,**
**Lu Yuan**[‡]**, Zicheng Liu**[‡]**, Zhangyang Wang**[§]
[†]Texas A&M University  [‡]Microsoft  [§]University of Texas at Austin
jiangziyu@tamu.edu, atlaswang@utexas.edu,
{yiche,mengcliu,dochen,xidai,luyuan,zliu}@microsoft.com

## Abstract

Recently, both Contrastive Learning (CL) and Mask Image Modeling (MIM) demonstrate that self-supervision is powerful to learn good representations. However, naively combining them is far from success. In this paper, we start by making the empirical observation that a naive joint optimization of CL and MIM losses leads to conflicting gradient directions - more severe as the layers go deeper. This motivates us to shift the paradigm from combining loss at the end, to *choosing the proper learning method per network layer*. Inspired by experimental observations, we find that MIM and CL are suitable to lower and higher layers, respectively. We hence propose to combine them in a surprisingly simple, "sequential cascade" fashion: early layers are first trained under one MIM loss, on top of which latter layers continue to be trained under another CL loss. The proposed *Layer Grafted* Pre-training learns good visual representations that demonstrate superior label efficiency in downstream applications, in particular yielding strong few-shot performance besides linear evaluation. For instance, on ImageNet-1k, Layer Grafted Pre-training yields 65.5% Top-1 accuracy in terms of 1% few-shot learning with ViT-B/16, which improves MIM and CL baselines by 14.4% and 2.1% with no bells and whistles. The code is available at https://github.com/VITA-Group/layerGraftedPretraining_ICLR23.git.

## 1 Introduction

Self-supervision has demonstrated undoubted power in learning strong visual representation, with two mainstream representative methods: Contrastive learning (CL) (Chen et al., 2020b; He et al., 2020; Chen et al., 2020d; 2021; Grill et al., 2020; Caron et al., 2021), and Mask Image Modeling (MIM) (Bao et al., 2021; He et al., 2021; Xie et al., 2022; Dong et al., 2021; 2022). The two methods follow different mechanisms, and often manifest different strengths. Generally, CL performs the instance-level task that pulls augmented views from the same image to be similar while pushing different images to distribute diversely, making it versatile at learning semantic-aware clustering structures across images. In contrast, MIM draws inspiration from BERT (Devlin et al., 2018) and performs masked token or pixel reconstruction that facilitates the learning of rich local structures within the same image. In particular, although the latter one, MIM, has recently surpassed CL on the fine-tuning performance of many datasets, CL often remains to be a top competitor in data-scarce, few-shot downstream applications (Chen et al., 2020c;d; Tian et al., 2020).

A natural question then follows: *are CL and MIM indeed complementary to each other, and is there a way to best combine their strengths?*. One immediate, conceptually simple idea is to refer to multiple task learning (MTL) and jointly optimize the two losses on top of the same backbone. Unfortunately, our preliminary experiment (See Section 2.2) shows that such a vanilla combination fails to improve over either baseline, in fact often compromising the single loss's performance. A deeper dive reveals that the two losses, when being optimized together, will incur increasingly severe

---

[*]Part of this work was conducted during a summer internship at Microsoft.

conflicts in the gradient directions, as the layers go deeper (see Figure 1). That causes considerable hurdles for the (pre-)training to effectively proceed.

We are then inspired to ask: *if the two losses conflict when both are placed at the end, how about placing them differently, such as appending them to different layers?* Based on experimental observations, it appears that lower layers tend to learn better from the MIM loss in order to capture local spatial details; while higher layers tend to benefit more from the CL loss in order to learn semantically-aware grouping and invariance. Inspired by so, we propose a simple MIM→CL Grafting idea to combine the bests of both worlds: (step i) first training the lower layers with MIM loss and fixing their weights, on top of which (step ii) higher layer weights continue to be trained under another CL loss. This simple cascaded training idea neatly separates MIM and CL losses to avoid their conflicts against each other if placed together; each loss is also strategically placed to pre-training its most suitable portion. Practically, we "'smooth out" the grafting by allowing lower layers to be slowly tuned in step ii. Our ablation experiments also find that **the order of grafting matters**, i.e., reversing MIM/CL loss locations and performing CL→MIM will considerably damage the performance. The contributions of this paper are summarized as follows:

- We propose Layer Grafted Pre-training, a principled framework to merge MIM and CL, improving representation learning beyond both, with no bells and whistles.

- We investigate the different preferences of lower and higher layers towards CL and MIM losses, and show the order of grafting to matter.

- Despite its embarrassing simplicity, the proposed Layer Grafted Pre-training demonstrates more desirable representation quality, and consequently superior label efficiency in downstream applications, yielding strong few-shot performance besides linear evaluation. For example, we achieve [65.5%, 77.8%, 77.7%] in terms of [1% few-shot, 10% few-shot, linear evaluation] performance, improving over MIM and CL baselines by [14.4%, 4.5%, 9.7%] and [2.1%, 2.4%, 1.0%], respectively.

## 2 METHOD

### 2.1 PRELIMINARY AND OVERVIEW

In Contrastive Learning (CL), the learning target is to pull the positive pairs together in the feature space while pushing negative pairs apart. Formally, the loss can be defined as:

$$\mathcal{M}(v_i, v_i^+, V^-, \tau) = \frac{1}{N} \sum_{i=1}^{N} - \log \frac{\exp\left(v_i \cdot v_i^+ / \tau\right)}{\exp\left(v_i \cdot v_i^+ / \tau\right) + \sum_{v_i^- \in V^-} \exp\left(v_i \cdot v_i^- / \tau\right)} \tag{1}$$

where $(v_i, v_i^+)$ represents features of the positive pairs while $(v_i, v_i^-)$ means features of negative pairs. Also, $V^-$ is the pool of negative features. $\tau$ denotes the temperature. $N$ is the number of samples. In practice, the positive pairs are often the different augmented views from the same image while the negative pool is composed by all the views from different images (Chen et al., 2021).

On the other hand, Mask Image Modeling (MIM) learns to reconstruct a corrupted image where some parts of the image or feature map are masked out. The learning target can be formulated as:

$$\mathcal{L}(x_i, M) = \frac{1}{N} \sum_{i=1}^{N} D(d(f(Mx_i)), x_i) \tag{2}$$

where $x_i$ and $M$ are input images and randomly generated masks, respectively. $f$ and $d$ represent the encoding and decoding functions, respectively. $d(f(Mx_i))$ is the generated image conditioned by masked image $Mx_i$. $D$ measures the difference between $d(f(Mx_i))$ and the original image $x_i$.

**Overview.** In the following parts of this section, we first introduce our preliminary exploration on the MTL of MIM and CL tasks in Section 2.2, which reveals the existence of the conflicting gradient direction. Afterward, in Section 2.3, we provide a simple separating idea towards mitigating the conflicts, which further leads to the proposed Layer Grafted Pre-training in Section 2.4.

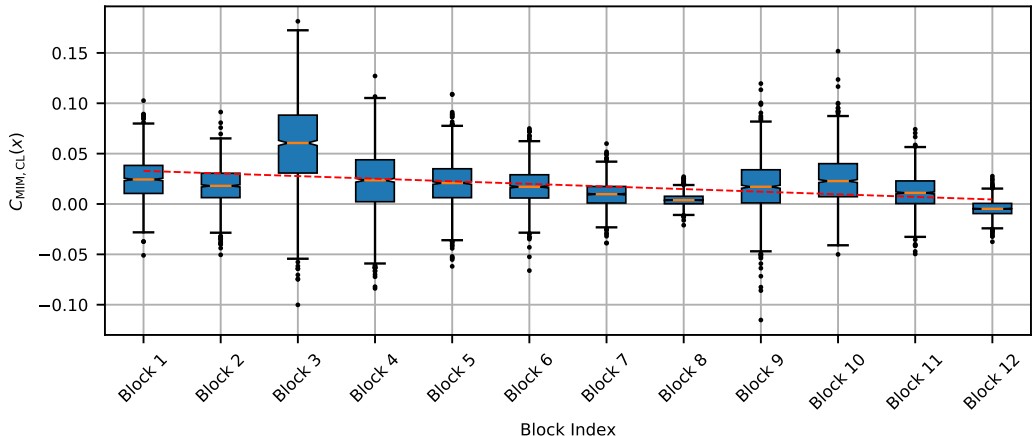

Figure 1: The box plot of $\mathbf{C}_{\text{MIM,CL}}(x)$ across different blocks for MTL combination of MIM and CL. This is measured on training datasets when the network is trained to 100 epochs (total 300 epochs). The red dash line indicates the linear regression of median numbers.

## 2.2 CONFLICTS PREVENT MULTI-TASK LEARNING FROM WORKING

Our first step towards answering the question of whether CL and MIM can complement each other is to examine the most straightforward and conceptually simple idea - Multi-Task Learning (MTL) combination. Specifically, each iteration of MTL is composed of two steps. Firstly, the images are augmented twice for computing the CL loss following Moco V3 Chen et al. (2021). Then, the image with minimal augmentation would be utilized for computing MIM loss following MAE He et al. (2021). These two losses share the same encoder and would be added together as the final loss.

As summarized in Table 1, MTL only yields a marginal performance improvement of 0.4% on linear evaluation compared to the MIM baseline. However, it is still much lower that the CL baseline (-8.3%). Moreover, on both 1% few-shot and fine-tuning performance, MTL is even inferior to both MIM and CL baselines. Similar observations were also made by Wang et al. (2022a).

We further conjecture that the conflicts between these two targets in the MTL combination is the cause of the bad performance. To verify it, we design a gradient surgery experiment by computing the cosine similarity between gradients of two tasks following Yu et al. (2020). Formally, the cosine similarity is calculated as follows:

$$\mathbf{C}_{\text{MIM,CL}}(x) = \frac{\nabla_\theta L_{\text{MIM}}(x)^T}{\|\nabla_\theta L_{\text{MIM}}(x)\|} \frac{\nabla_\theta L_{\text{CL}}(x)}{\|\nabla_\theta L_{\text{CL}}(x)\|} \tag{3}$$

where $L_{\text{MIM}}$ and $L_{\text{CL}}$ denote the losses for MIM and CL, respectively. $x$ is a batch of input samples. We measure the distribution of $\mathbf{C}_{\text{MIM,CL}}(x)$ across different layers of a pre-trained MTL model. As shown in Figure 1, there always exist negative values for $\mathbf{C}_{\text{MIM,CL}}(x)$, where the MIM and CL are optimized in opposite directions. Moreover, the gradient direction varies across layers - more severe as layers go deeper.

Also, the conflicts can be reflected in two losses' contradictory targets to enforce. The MIM loss, for instance, requires that the reconstruction have the same brightness, color distribution, and positions as the input image, therefore the model needs to be sensitive to all these augmentations. Conversely, CL loss is designed to ensure that the model remains invariant regardless of different augmentations.

## 2.3 ADDRESSING THE CONFLICTS VIA SEPARATING

Given the conflicts of the MTL combination, we ask the following question: *if the two losses conflict when both are placed at the end, how about placing them differently, such as appending them to different layers?* Fortunately, recent empirical evidence suggests that CL and MIM may favor different pre-training methods. For MIM, Wang et al. (2022c) points out that, when only the pre-trained lower

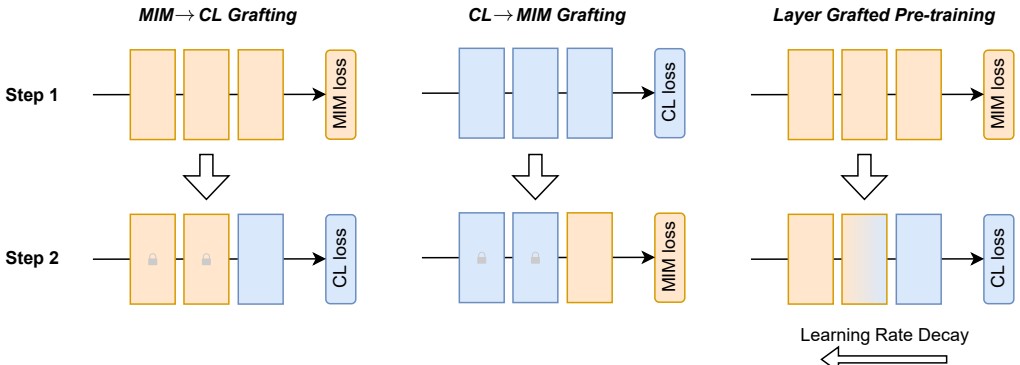

Figure 2: The pipelines of the MIM→CL, CL→MIM Grafting, and Layer Grafted Pre-training. The former two are employed for preliminary experiments. The latter one is the final adopt pipeline, which is the 'smooth out' version of MIM→CL Grafting.

Table 1: Illustration of preliminary study experiments' performance on ViT-B/16. Linear, 1% and Fine-tuning denote linear evaluation, 1% few-shot and fine-tuning performance, respectively. The performance of MIM and CL are from MAE (He et al., 2021) and MocoV3 (Chen et al., 2021), respectively. MTL combination denotes the Muti-Task Learning (MTL) Combination of MIM and CL. MTL combination is pretrained for 300 epochs. For step 1 of MIM→CL and CL→MIM Grafting, we directly adopt the pre-trained model of MAE and MoCo V3, respectively. Step 2 of MIM→CL and CL→MIM Grafting is trained for 100 epochs.

| Method | Linear | 1% | Fine-tuning |
| --- | --- | --- | --- |
| MIM (MAE) | 68.0 | 51.1 | 83.6 |
| CL (Moco V3) | 76.7 | 63.4 | 83.2 |
| MTL combination | 68.4 | 47.6 | 81.0 |
| CL→MIM Grafting | 65.5 | 32.5 | 82.5 |
| MIM→CL Grafting | 74.5 | 56.5 | 83.6 |

layers are retained while the higher layers are reset to random initialization, most of the gain is still preserved for downstream fine-tuning tasks. Based on this observation, the lower layers appear to be a key element in MIM. On the other hand, Chen et al. (2021) finds CL to be ineffective and even unstable for training the projection layer, the earliest layer of ViT (Dosovitskiy et al., 2020). Fixing its weight to be random initialization can even yield significantly higher performance. Additionally, CL excels at semantic concepts, which happen often at higher layers of the neural network.

Driven by the above analysis, we propose a simple MIM→CL Grafting framework. As shown in Figure 2, MIM→CL Grafting can be separated into two steps: (step i) the lower layers are first trained with MIM and then fixed, on the top of which (step ii) higher layers continue to learn with CL. Despite the simplicity of this framework, it yields promising preliminary results as shown in Table 1, exceeding the MTL combination by [6.1%, 8.9%, 2.6%] for [linear evaluation, 1% few-shot, Fine-tuning] performance, respectively. In contrast, when the order of two tasks is reversed, the resulting CL→MIM Grafting, would suffer a dramatic drop in performance, which is even lower than the MTL combination by [2.9%, 15.1%] in terms of [linear evaluation, 1% few-shot] performance, respectively. The huge gap between CL→MIM and MIM→CL Grafting further confirms the preference of MIM and CL towards lower and higher layers, respectively.

The example discussed at the end of Section 2.2 can also explain why this preference difference happens: The two different prior knowledge types requested by MIM and CL, while seemingly at odds, may work together at different layers of the model. For example, the sensitivity to augmentations can be helpful for recognizing the local feature with strong color patterns (Xiao et al., 2020) in the lower layers (i.e. the fur of leopards). Meanwhile, for a consistent semantic understanding,

the influence of the lightness difference should be eliminated when it comes to understanding the context feature at higher layers.

## 2.4 LAYER GRAFTED PRE-TRAINING

To fully unleash the power of Grafting, we 'smooth out' the boundary of MIM→CL grafting to avoid a sudden change in the feature space. Specifically, rather than fixing the lower layers, we assign them a small learning rate. The resultant method, termed as Layer Grafted Pre-training, is shown in Figure 2. In Section 3.3, we also explore other LR choices and our results indicate that employing small and large LR for lower and higher layers, respectively, yields the best performance.

By effectively capturing the augmentation-sensitive features (i.e. the colors) in the lower layers with MIM while learning semantic alignment in the higher layers with CL. The proposed Layer Grafted Pre-training enables the learning of strong visual representations. It not only provides strong inter-class variance that helps to cluster but also benefits the intra-class variance by keeping the diversity of samples in the early features.

## 3 EXPERIMENT

### 3.1 SETTING

**General.** We conduct all the experiments on ImageNet-1k (Deng et al., 2009) with Nvidia V100 GPUs. The code is implemented with Pytorch (Paszke et al., 2019).

**Backbone.** We adopt the standard ViT-B and ViT-L architecture (Dosovitskiy et al., 2020) with the token size of $16 \times 16$. The ViT-B is by default employed unless specified. When training with CL loss, we employ the projection and prediction head following Moco V3 (Chen et al., 2021). The settings for pre-training and evaluation protocols can be found at Appendix A.5.

### 3.2 COMPARISON WITH STATE-OF-THE-ART METHODS

We start by verifying the effectiveness of the proposed Layer Grafted Pre-training by comparing it with state-of-the-art methods. As shown in Table 2, in ViT-B/16, compared to the employed MIM and CL baselines, the proposed Layer Grafted Pre-training leads to a consistent improvement. For instance, it improves MAE and Moco V3 by [9.7%, 14.4%, 4.5%] and [1.0%, 2.1%, 2.4%] for [linear evaluation, 1% few-shot, 10% few-shot], respectively.

Compared to close competitors which also attempt to combine MIM and CL, the proposed Layer Grafted Pre-training surpasses iBoT by 1.5% for linear evaluation performance. Compared to SIM, the proposed Layer Grafted Pre-training yields an improvement of 1.1% and 0.2% for linear evaluation and 1% few-shot learning performance, respectively.

Table 2: Comparison with State-of-The-Arts on ViT-B/16 and ViT-L/16. Linear, 1% and 10% denote the top-1 accuracy (%) of linear evaluation, 1% and 10% few-shot learning, respectively. †: We employ the result of iBoT without augmentations from Zhou et al. (2021) for fair comparison.

| BackBone | Method | Linear | 1% | 10% |
|---|---|---|---|---|
| ViT-B/16 | MAE (He et al., 2021) | 68.0 | 51.1 | 73.3 |
| | Moco V3 (Chen et al., 2021) | 76.7 | 63.4 | 75.4 |
| | iBoT† (Zhou et al., 2021) | 76.0 | - | - |
| | SIM (Tao et al., 2022) | 76.4 | 65.3 | - |
| | C-MAE (Huang et al., 2022) | 73.9 | 65.3 | 77.3 |
| | MimCo (Zhou et al., 2022) | 70.2 | 62.7 | - |
| | Layer Grafted Pre-training (Ours) | **77.7** | **65.5** | **77.8** |
| ViT-L/16 | MAE (He et al., 2021) | 75.8 | 55.2 | 78.7 |
| | Moco V3 (Chen et al., 2021) | 77.6 | - | - |
| | Layer Grafted Pre-training (Ours) | **81.0** | **69.3** | **80.1** |

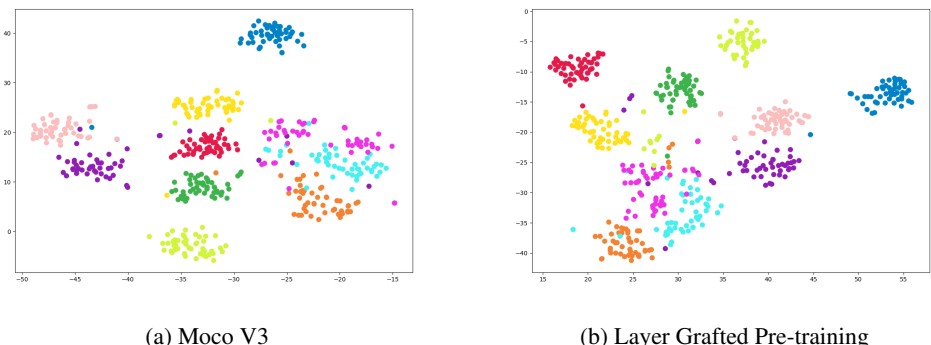

(a) Moco V3

(b) Layer Grafted Pre-training

Figure 3: t-SNE (Van der Maaten & Hinton, 2008) visualization for feature distribution of Moco V3 and Layer Grafted Pre-training. Different colors represent different classes. Best viewed in color.

Our method also demonstrates good scalability toward larger models size. For instance, when scaling from ViT-B/16 to ViT-L/16, Layer Grafted Pre-training further improves accuracy by [3.3%, 3.8%, 2.3%] in terms of [linear evaluation, 1% few-shot, 10% few-shot], respectively. Remarkably, the gap above Moco V3 in linear evaluation performance also increases from 1.0% to 3.4%.

We further qualitatively evaluate the representations learned by the proposed Layer Grafted Pre-training using t-SNE (Van der Maaten & Hinton, 2008). As shown in Figure 3b, the proposed Layer Grafted Pre-training shows better inter-class variance. For example, the categories represented by pink (•) and light blue (•) points are hard to separate given they are very close with each other in Figure 3a. In contrast, for representation of the proposed Layer Grafted Pre-training, they form two clusters with a clear boundary in Figure 3b. Besides, the proposed Layer Grafted Pre-training also

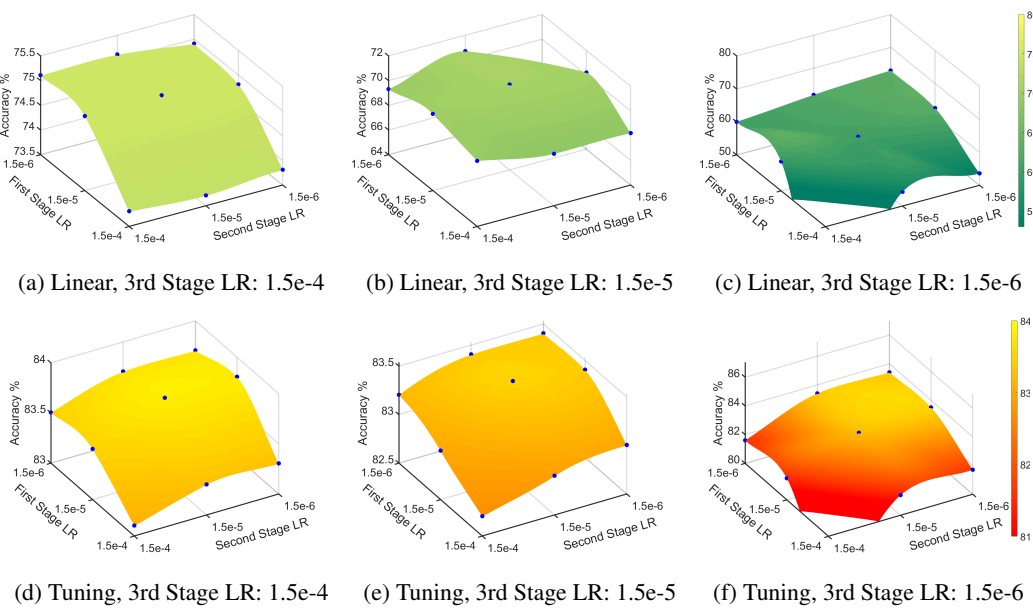

(a) Linear, 3rd Stage LR: 1.5e-4   (b) Linear, 3rd Stage LR: 1.5e-5   (c) Linear, 3rd Stage LR: 1.5e-6

(d) Tuning, 3rd Stage LR: 1.5e-4   (e) Tuning, 3rd Stage LR: 1.5e-5   (f) Tuning, 3rd Stage LR: 1.5e-6

Figure 4: Illustration of the LR grid search results for different stages in terms of linear evaluations and fine-tuning performance. The grid is [1.5e-6,1.5e-5,1.5e-4] for each stage. [(a), (b), (c)] and [(d), (e), (f)] denotes the linear evaluation and fine-tuning performance with third stage LR of [1.5e-4, 1.5e-5, 1.5e-6], respectively. [(a), (b)] and [(e), (f)] share the same color bar with (c) and (g), respectively. That of (d), (e) and (f) are also the same. The tested points are highlighted with blue dots in each plot. Best view in color.

shows better intra-variance: the red (•), green (•) and yellow (•) points of Moco V3 collapse to a smaller region than the proposed Layer Grafted Pre-training.

### 3.3 LR search Layer Grafted Pre-training

We further verify if small and large LR work the best for the proposed Layer Grafted Pre-training. Specifically, we study the performance with different LR settings on the three stages on ViT-B/16 (Refer to fine-tuning part of Appendix A.5 for the definition of stages), where each stage is searched with a grid of [1.5e-6,1.5e-5,1.5e-4]. As demonstrated in Figure 4, when the LR increase from 1.5e-6 to 1.5e-4 for the third stage, both linear evaluation and fine-tuning performance achieve an improvement. Taking linear evaluation as an example, as shown in Figure 4a, 4b and 4c, when the LR of the third stage increase from 1.5e-6 to 1.5e-5 and 1.5e-4, the performance range could improve from 2.5%-63.0% to 64.8%-70.9% and 73.7%-75.1%, respectively. In contrast, a big LR of the first stage would lead to a drop in performance. For instance, in terms of the linear evaluation performance with third stage LR of 1.5e-4, as shown in Figure 4a, the performance ranges are 74.9%-75.1%, 74.8%-75.0% and 73.7%-73.8% for first stage LR of 1.5e-6, 1.5e-5 and 1.5e-4, respectively. The LR of the second stage is not as sensitive as that of the first or third stage.

The trend of fine-tuning performance is also similar to that of the linear evaluation performance. The preference for larger LR for higher layers indicates that they can benefit by performing CL. Meanwhile, lower layers prefer a smaller LR means that keeping MIM features for these layers can be helpful.

### 3.4 More Ablations

**Fine-tuning Performance Comparison.** We also compare State-of-The-Art methods for fine-tuning performance. As shown in Table 3, the proposed Layer Grafted Pre-training yields a competitive fine-tuning performance of 83.9%, which is 0.3% and 0.7% higher than the employed MIM (MAE) and CL (Moco V3) baselines, respectively. Moreover, Layer Grafted Pre-training also surpasses SIM by 0.1%.

**Partial Fine-tuning.** Follow MAE (He et al., 2021), we evaluate the performance of the proposed Layer Grafted Pre-training with different number of fixing blocks. As

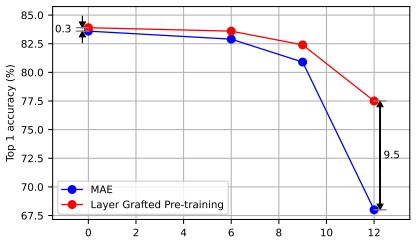

Figure 5: Comparison between the proposed Layer Grafted Pre-training and MAE under different numbers of fixing blocks on ViT-B/16 in terms of fine-tuning performance. The training dataset is the full ImageNet-1k. Best view in color.

illustrated in Figure 5, Layer Grafted Pre-training consistently yields higher performance than MAE. And this gap continue to grow larger when more layers are fixed, indicating the superiority of the representations learned by the proposed method.

**Variance-Invariance-Covariance Analysis.** A study of the Variance-Invariance-Covariance pattern for the output of each block is conducted in order to better understand the Layer Grafted Pre-training. As illustrated in Figure 6, we find that the VIC pattern of Layer Grafted Pre-training tends

Table 3: Top 1 Fine-tuning performance comparison.

| BackBone | Method | Fine-tuning |
|---|---|---|
| ViT-B/16 | MAE (He et al., 2021) | 83.6 |
| | Moco V3 (Chen et al., 2021) | 83.2 |
| | SIM (Tao et al., 2022) | 83.8 |
| | ConMIM (Yi et al., 2022) | 83.7 |
| | Layer Grafted Pre-training (Ours) | **83.9** |
| ViT-L/16 | MAE (He et al., 2021) | **85.9** |
| | Moco V3 (Chen et al., 2021) | 84.1 |
| | Layer Grafted Pre-training (Ours) | **85.9** |

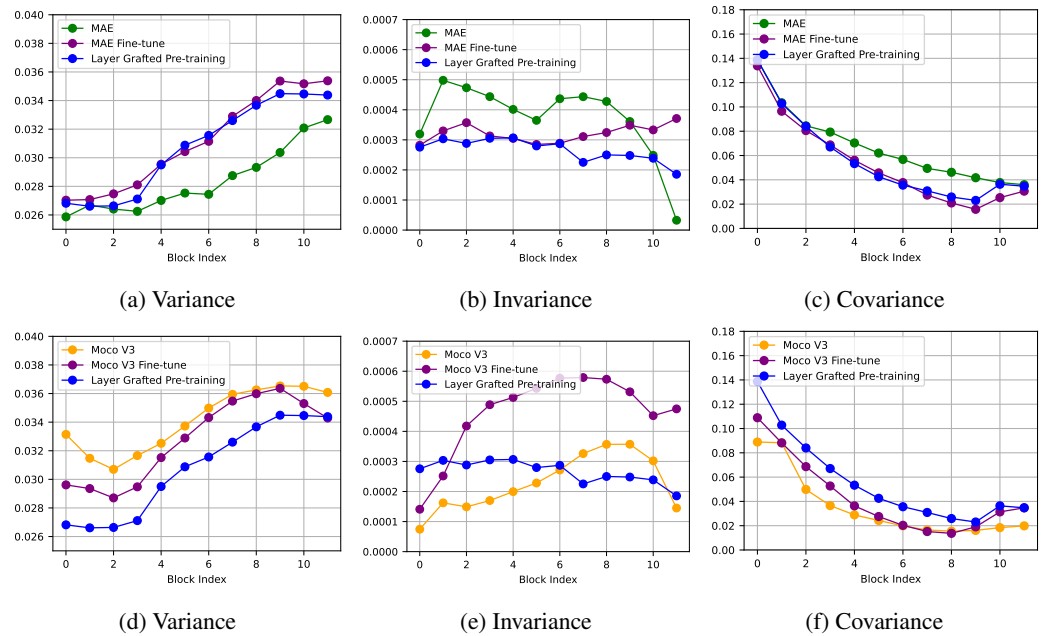

Figure 6: The Variance-Invariance-Covariance (VIC) analysis for different methods. VIC are computed on ViT-B/16 following Bardes et al. (2021). The input features are first averaged over all tokens and then normalized to remove the effect from magnitude. [(a), (d)], [(b), (e)] and [(c), (f)] study variance, covariance and invariance, respectively. Best view in color.

Table 4: Layer Grafted Pre-training on ViT-B/16 with VICReg (Bardes et al., 2021). We train ViT-B/16 with VICReg for 100 epochs as the baseline. For Layer Grafted Pre-training - VICReg, the CL loss of stage ii is replaced with VICReg loss.

| Method | Linear | Fine-tuning |
|---|---|---|
| VICReg (Bardes et al., 2021) | 70.1% | 81.2% |
| Layer Grafted Pre-training - VICReg (Ours) | 74.9% | 83.6% |

to be similar to that of fine-tuning. In the MAE case, the VIC curve of Layer Grafted Pre-training closely matches that of MAE Fine-tune, much closer than MAE itself. The similarity between the proposed Layer Grafted Pre-training and fine-tuning in the VIC pattern also explains the high few-shot performance: weights of the pre-training do not need to change substantially to match the downstream task.

**Layer Grafted Pre-training with VICReg.** We further examine the generalizability of the proposed idea on a different pre-training method - VICRegn (Bardes et al., 2021). As shown in Table 4, when replacing the CL loss with the VICReg loss, the proposed Layer Grafted Pre-training still yields strong performance, surpassing the VICReg baseline by [4.8%, 2.4%] for [linear evaluation, fine-tuning] performance, respectively.

# 4   RELATED WORKS

**Contrastive Learning.** CL performs instance classification tasks by contrasting positive pairs against negative pairs (Chen et al., 2020b; He et al., 2020; Zhuang et al., 2019; Dosovitskiy et al., 2014). Other close works also explore learning without negative samples (Grill et al., 2020; Bardes et al., 2021; Caron et al., 2021; Zbontar et al., 2021; Chen & He, 2021) and the clustering based approachs (Caron et al., 2020).

One common merit of these methods is their strong performance on learning good representations, which shows strong clustering pattern (Dwibedi et al., 2021) and leads to state-of-the-art few-shot/semi-supervised performance (Chen et al., 2020c; Tian et al., 2020; Li et al., 2021; Jiang et al., 2022; You et al., 2022). However, they contain an implicit assumption that the features should be invariant to heavy augmentations, which, however, could further lead to worse performance when the downstream performance violates it (Xiao et al., 2020). The proposed Layer Grafted Pre-training address this via leveraging MIM for processing the features of lower layers.

**Mask Image Modeling.** Mask Image Modeling (MIM) is inspired by the success of BERT (Devlin et al., 2018) in Natural Language Processing (NLP). iGPT (Chen et al., 2020a) begins the exploration of this idea in Computer Vision (CV). The emergence of ViT (Dosovitskiy et al., 2020) further shrinks the gap of backbones between CV and NLP, motivating more researchers to delve into this direction. Beit (Bao et al., 2021; Peng et al., 2022), MaskFeat (Wei et al., 2022) and Peco (Dong et al., 2021) focus on predicting tokens. MAE (He et al., 2021) and simMIM (Xie et al., 2022) further show the possibility of directly reconstructing the original pixels. Following works (Dong et al., 2022; Chen et al., 2022; Wang et al., 2022b) continue to improve the performance or extend to other modalities. However, MIM achieves the most success in fine-tuning with enough data points. For downstream tasks with limited data, MIM fails to surpass CL given the lack of linear separability for its representations (Tao et al., 2022). We address this drawback by employing CL for learning the semantic alignment for higher layers.

**Bridging Contrastive Learning and Mask Image Modeling.** Only recently, researchers begin to explore the potential of combining MIM and CL. iBoT (Zhou et al., 2021), one of the pioneers in this direction, proposes to switch the modeling of images to the modeling of features. Some concurrent works also follow this self-distillation paradigm (Tao et al., 2022; Assran et al., 2022). However, just like CL, this paradigm still relays on the involvement of strong augmentations to avoid collapsing, which could lead to over-suppressing of some features (i.e. color) (Xiao et al., 2020). In contrast, we employ MAE (He et al., 2021), an image modeling framework that is free from strong augmentations. Besides, previous combination works treat the network as a whole while we provide a new layer-wise perspective.

**Comparison to other multiple-step pre-training tasks.** One may relate the proposed method with previous multiple-step pre-training tasks like intermediate fine-tuning ( e.g., finetuning a MIM model using ImageNet22k and transfer to ImageNet1k (Bao et al., 2021)) or self-supervised fine-tuning like Reed et al. (2022). The main differences lie in two aspects: (1) The key innovation of the proposed method is to reveal and utilize the layerwise difference between MIM and CL. In contrast, intermediate finetuning and self-supervised fine-tuning are treating the model as a whole; and (2) While intermediate finetuning and self-supervised are designed for the **same pretraining methods** across **different domains**, the proposed method is devised for **different pretraining methods** in the **same domain**.

## 5 CONCLUSION

In this work, we propose Layer Grafted Pre-training, a simple yet principled method for understanding and bridging two popular types of self-supervised learning methods - Mask Image Modeling (MIM) and Contrastive Learning (CL). Our work provides a simple remedy to the conflicts between MIM and CL and further reveals the different preferences of these two methods toward different parts of the neural network. It advances the quality of the self-supervised representations and achieves strong performance on linear evaluation and few-shot performance. Potential future work includes assessing or extending the proposed method to real-world unlabeled data with more challenges such as long-tail distribution or imbalance Jiang et al. (2021).

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

Table 5: Comparison between MIM (MAE) and CL (Moco V3) for frozen features from blocks [6,9,12], on top of which we employ various numbers of blocks (#Blocks) for fine-tuning. 0 block indicates that only a linear classification head is employed (identical to linear evaluation). Top 1 accuracy on ImageNet-1K is reported and the best performance under each setting is highlighted with bold text.

| #Blocks | The block index of the feature | | | | | |
| | 6 | | 9 | | 12 | |
| | MIM | CL | MIM | CL | MIM | CL |
|---|---|---|---|---|---|---|
| 0 | 38.9% | **43.6%** | 59.3% | **65.5%** | 68.0% | **76.7%** |
| 1 | 70.1% | **72.7%** | 77.8% | **78.1%** | 78.9% | **79.1%** |
| 2 | 76.3% | **76.8%** | **80.2%** | 79.2% | **80.6%** | 79.7% |
| 4 | **78.5%** | 78.1% | **81.2%** | 79.4% | **81.4%** | 79.2% |

## A  APPENDIX

This appendix contains the following details that we could not include in the main paper due to space restrictions.

### A.1  MORE ANALYSIS FOR THE LAYER-WISE DIFFERENCE BETWEEN MIM AND CL

We provide more analysis to further understand the layerwise difference between MIM and CL. We start by analyzing the average attention distance across different layers. As shown in Figure 7, on the one hand, for the deep layers (i.e. from 8th to 12th blocks), the average attention distance of CL would keep increasing, where the aggregation of local features is likely to happen. In contrast, the average attention distance of MIM keeps the same level for the deep layers. On the other hand, the average attn distance of shallow layers (i.e. 1st and 2nd blocks) of CL is much larger than that of MIM, which may distract the model from extracting local features. The proposed method combines the lower and higher layers' patterns of MIM and CL, respectively, forming a gradually increasing attention distance pattern. Remarkably, this pattern echos the philosophy of gradually increasing receptive field for designing network architecture He et al. (2016); Liu et al. (2021).

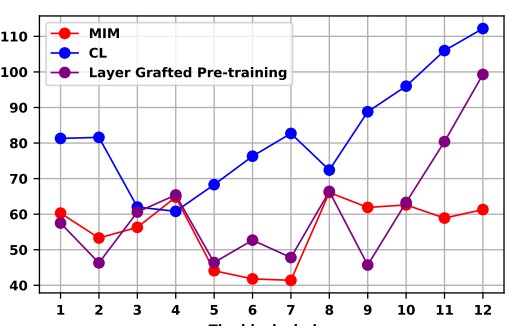

Figure 7: Demonstration of average attention distance (Dosovitskiy et al., 2020) for MIM (MAE), CL (Moco V3) and Graft in terms of the across all attention heads.

Secondly, we study the different properties of features across different layers. Specifically, in ImageNet 1K, we turn several random initialized blocks on top of features from different layers. As demonstrated in Table 5, when only fine-tuning the classification head (#Block = 0), the performance of MIM is much lower than CL, indicating that the feature of CL is more close to semantic representation. By contrast, when increasing the turnable blocks, the performance of MIM would significantly increase and even surpass CL, demonstrating the features of MIM better encode the local features. The potential of these local features can be stimulated by adding enough modules to aggregate them. The proposed Layer Grafted Pre-training employs MIM for producing high-quality early features while utilizing the CL in higher layers for aggregation.

### A.2  DISCUSSION AND COMPARISON WITH CONCURRENT WORKS

In this section, we discuss the difference between the proposed method with four concurrent works (Tao et al., 2022; Huang et al., 2022; Zhou et al., 2022; Yi et al., 2022) and provide more

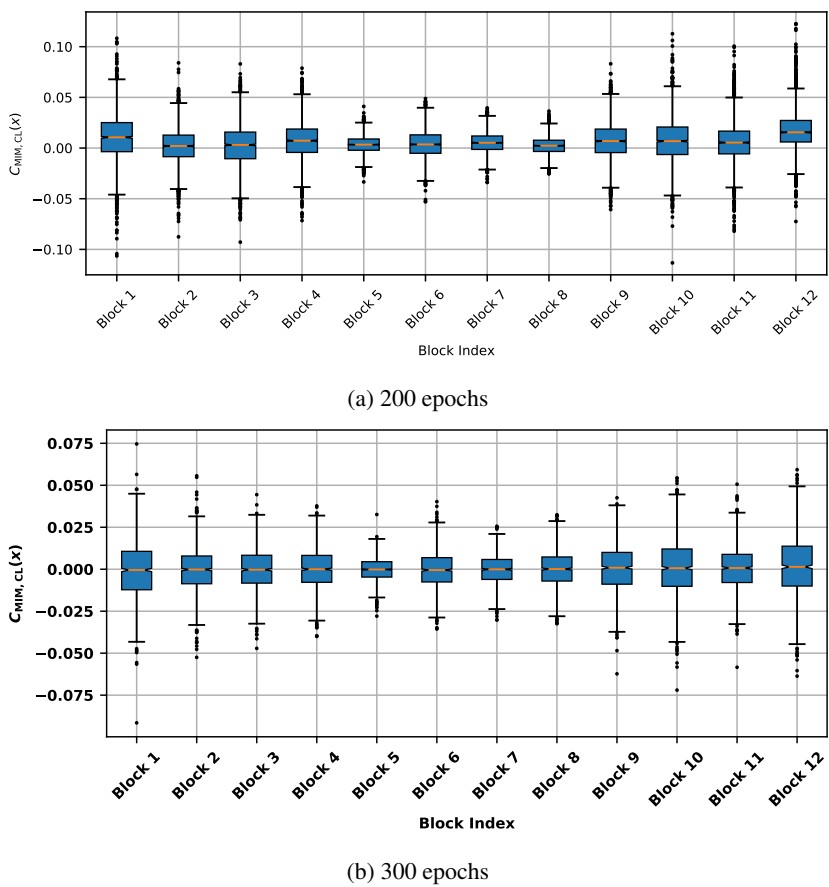

(a) 200 epochs

(b) 300 epochs

Figure 8: The box plot of $\mathbf{C}_{\mathrm{MIM,CL}}(x)$ across different blocks for MTL combination of MIM and CL. This is measured on training datasets when the network is trained to (a) 200 epochs and (b) 300 epochs (total 300 epochs). The red dash line indicates the linear regression of median numbers.

comparisons. To combine the strength of MIM and CL, SIM (Tao et al., 2022) proposes to predict the dense representations of an augmented view for enforcing semantic alignment and spatial sensitivity simultaneously. CMAE (Huang et al., 2022) proposes two new components: pixel shift and feature decoder. MimCo (Zhou et al., 2022) utilizes the CL pre-trained model as a teacher and performs patch-level and image-level reconstruction tasks. ConMIM (Yi et al., 2022) utilizes contrastive constraints to produce a dynamic masked prediction target. The difference between the proposed Layer Grafted Pre-training and these works are as follows:

- While all the concurrent works are treating the network as a whole, we reveal the different preferences of MIM and CL towards their internal different layers, which motivates us a design a novel layerwise method.

- Our method employs the original design of MIM and CL, which not only is simple but also enables an apple-to-apple comparison. In contrast, It's non-straightforward to tell whether the improvements of the concurrent works are from the newly introduced module or the original MIM/CL design.

- The proposed Layer Grafted Pre-training provides an in-depth analysis of the reason why MIM and CL cannot be directly combined together through gradient analysis.

Also, here we further analyze why the proposed method fails to surpass the concurrent work C-MAE (Huang et al., 2022) in terms of fine-tuning performance. One possible reason lies in whether the masked view is employed for contrastive learning. C-MAE is contrasting a masked and a full image while the proposed method is contrasting two full images following Moco V3. The difference in design further leads to different strengths: On the one hand, empirical results highlight that the

Table 6: ADE20K semantic segmentation comparison using UperNet with different pre-training methods.

| Method | mIoU |
|---|---|
| MAE (He et al., 2021) | 48.1 |
| Moco V3 (Chen et al., 2021) | 47.3 |
| Layer Grafted Pre-training (Ours) | **48.7** |

masked view can benefit the downstream fine-tuning task (Touvron et al., 2022; He et al., 2021), which may be because it helps to learn the correlation between sparse patches that cannot be built under full view. On the other hand, contrasting full images leads to a smaller gap with downstream tasks and thus benefiting the downstream few-shot and linear evaluation tasks.

### A.3 MORE GRADIENT ANALYSIS RESULTS

We measure the distribution of $\mathbf{C}_{\text{MIM,CL}}(x)$ across different training epochs and confirmed that the statistical results persist the same across the entire training process, rather than just a few specific epochs. As two examples, Figures 8a and 8b show that a considerable part of values is negative in the 200 and 300 epochs.

### A.4 TRANSFER LEARNING RESULTS

We further evaluate the proposed method for the standard transfer semantic segmentation task. As shown in Table 6, on ADE20K with UperNet, the proposed method achieves higher performance than both MAE and Moco V3.

### A.5 MORE SETTINGS

**Pre-training.** We by default adopt MAE (He et al., 2021) and Moco V3 (Chen et al., 2021) as our MIM and CL frameworks, respectively. For the first step of Layer Grafted Pre-training, since it identically follows the original MIM pipeline, we directly adopt the pre-trained model of MAE (He et al., 2021) and conduct the second step, where we initialize the network with the MIM pre-trained model and train with Moco V3 (Chen et al., 2021) for 300 epochs. For LR, We first split the network into three stages. Each of them contains the same number of blocks. (i.e. 4 and 8 blocks for ViT-B and ViT-L, respectively.) Then, the base LR of the first and second stages (corresponding to lower layers) is assigned as 1.5e-5 while the third stage is set as 1.5e-4 by default. In the second stage of ViT-L, we further ensure the early layers of the resultant model are close to the MIM pre-training by minimizing the $l^2$ distance between the first 12 layers between them (Refer Section A.6 for more ablations). Other settings are identically followed from Moco V3 (Chen et al., 2021).

**Fine-tuning.** For fine-tuning, we train with AdamW (Loshchilov & Hutter, 2017) for 100 epochs following MAE (He et al., 2021). We employ a base LR of 5e-4 with linear scaling rule (Goyal et al., 2017). The layer-wise LR decay ratio is set as 0.6 (Clark et al., 2020). For other settings such as data augmentation, LR scheduling or weight decay, we identically follow He et al. (2021).

**Few-shot Learning.** We conduct few-shot learning with 1% or 10% available labels. The subsampling splits are adopted from Chen et al. (2020c). For 1% few-shot evaluation, following Caron et al. (2021), we first generate frozen features on training images without data augmentation, on top of which a logistic regression classifier is trained for prediction. For 10% semi-supervised learning, we train from the first layer of the projection head following Chen et al. (2020c). We train for 400 epochs with an initial base LR of 3e-5. Other settings are identical to the fine-tuning.

**Linear Evaluation.** For linear evaluation, we train a linear classifier on top of frozen pre-train features to measure the quality of the visual representations following common practices (Chen et al., 2020b). Following Moco V3 (Chen et al., 2021), the classifier is trained for 90 epochs with SGD optimizer and weight decay of 0. The LR is swept for each case.

Table 7: Ablation study for $l^2$ regularization on ViT-L. Top 1 accuracy on ImageNet-1K is reported and the best performance under each setting is highlighted with bold text.

| $l^2$ regularization | Linear | 1% | 10% | Finetune |
|:---:|:---:|:---:|:---:|:---:|
| ✗ | 80.5 | 68.9 | 79.8 | 85.7 |
| ✓ | **81.0** | **69.3** | **80.1** | **85.9** |

## A.6 ABLATION STUDY FOR $l^2$ REGULARIZATION

In this section, we ablation study the effectiveness of $l^2$ regularization mentioned at section A.5. As shown in Table 7, employing the $l^2$ regularization can help to preserve the Mask Image Modeling features of the lower layers and yield consistent improvement across multiple benchmarks.

