# OpenReview forum: "Layer Grafted Pre-training: Bridging Contrastive Learning And Masked Image Modeling For Label-Efficient Representations"
_ICLR.cc/2023/Conference — ICLR 2023 poster_

### Official Review · Reviewer_nkwf · 2022-10-23

**Confidence:** 4
**Correctness:** 3
**Technical Novelty And Significance:** 3
**Empirical Novelty And Significance:** 3
**Recommendation:** 6

**Clarity, Quality, Novelty And Reproducibility:**

Clarity
- Clear, only minor flaws

Quality
- Technically strong results

Novelty
- Minor variations to existing techniques. But its empirical finding is valuable to be shared

Reproducibility
- Detailed configurations are provided in the manuscript.

**Strength And Weaknesses:**

Strengths
- The writing is clear and easy to understand.
- This manuscript tackles a recently popular and interesting problem that unifies two large paradigms (i.e., Contrastive Learning (CL) and Masked Image Modeling (MIM)) that divide recent self-supervised learning.
- The proposed method outperforms the state-of-the-art self-supervised learning baselines on the image-classification tasks.

Weaknesses
- Overall, the authors validate the proposed method on CL-centric evaluation metrics (e.g., linear evaluation, few-shot classification). One of the main differences between CL and MIM is that "MIM helps the model to learn rich local structures within the image" (as mentioned in the manuscript), which can be beneficial to dense prediction-type downstream tasks such as object detection and semantic segmentation. For example, MAE [1] has shown superiorities of MIM in COCO object detection and segmentation, and on ADE20K semantic segmentation, while it shows poor performance on linear probing compared to fine-tuning on image classification tasks. However, the proposed method only highlights the benefits of image classification tasks, which are close to the benefits of CL. Moreover, improvements compared to CL are much smaller than MIM. In short, my question is, " Is the proposed method a unified method of CL and MIM or a slightly better variant of CL?" If so, does the proposed method inherit the benefits of MIM? (e.g., dense prediction tasks)
- The key observation of the manuscript is that CL and MIM loss objectives are in conflict. However, I am questionable that the proposed MTL combination is the right way to conclude it. Specifically, the authors enforce the model to discriminate against all the augmentations under MIM loss, while invariant against the same augmentations under CL loss. I think combining MIM without augmentation (or minimal) and CL with augmentation is a more straightforward approach with less conflict. Furthermore, in the experiments section, the authors also adopt MAE models, which are pre-trained without augmentations. I would like to ask the authors to provide more explanations and valuable insights about this.
- In the preliminary section 2.1, it is unclear which features are used for contrastive learning; which are they between outputs of the encoding function $f$ or the decoding function $d$? Does the decoding function $d$ be discarded after training MIM?

[1] He et al., "Masked Autoencoders Are Scalable Vision Learners," CVPR 2022

**Summary Of The Paper:**

This manuscript proposes Layer Grafted Pre-training, a simple two-step approach for bridging two recent self-supervised learning methods, which are Masked Image Modeling (MIM) and Contrastive Learning. The key observation is that both objectives have conflict aspects, and separating them into lower and higher layers can enhance the stability of joint optimization of MIM and CL. Specifically, in a cascade fashion, the proposed method pre-trains early layers under MIM loss and then trains the other layers under CL loss. The extensive experiments showed the effectiveness of the proposed method on image classification tasks such as few-shot classification and linear evaluation.

**Summary Of The Review:**

Overall, I have several concerns, as I mentioned above weaknesses section.
For now, I weakly recommend accepting this manuscript.

---

### Official Review · Reviewer_Yr9u · 2022-10-24

**Confidence:** 4
**Clarity, Quality, Novelty And Reproducibility:** The paper has good clarity, limited n…
**Correctness:** 3
**Technical Novelty And Significance:** 2
**Empirical Novelty And Significance:** 2
**Recommendation:** 5

**Strength And Weaknesses:**

Strength:
- Combining contrastive learning and masked image modeling is a popular problem and has attracted much attention. This paper provides a very simple solution to benefit from contrastive learning and masked image modeling at the same time.
- The paper shows high quality in writing. The presentation of both methods and experiments is intuitive. It is easy to follow and comprehend.

Weakness:
- The paper presents limited novelty and contribution. The method simply finetunes a strong MIM pretrained model using contrastive learning, and then transfer it to downstream classification task. It can be viewed as an intermediate fine-tuning, e.g.finetune a MIM model using ImageNet22k and transfer to ImageNet1k,  which is used in BEiT and adopted by some follow-up papers. The intermediate fine-tuning has been demonstrated to be very effective and can bring 1.5% Top-1 accuracy.
- The experiments are insufficient to demonstrate the effectiveness. The method only improves 0.3% top-1 accuracy when fine-tuning on ImageNet. In addition, fine-tuning experiments are only conducted using ViT-B/16 backbone. The results of ViT-L/16 should be reported.
- The experiments in Table 4 show that the method doesn't work when using VICReg. The fine-tuning result is the same as the original MAE, i.e., 83.6%.
- What about conducting multi-task learning but performing masked image modeling at lower layers? This is an important baseline according to your motivation.
- There are some papers (e.g., A, B, C, etc) that also combine contrastive learning and masked image modeling, including a few listed in related work. More comparison and discussion can be added.
- The paper can also be viewed as using self-supervised pretraining (CL) improves self-supervised pertaining (MIM), which is similar to [D]. Discussion can be added.

[A] Siamese image modeling for self-supervised vision representation learning.

[B] MimCo: Masked Image Modeling Pre-training with Contrastive Teacher

[C] Masked Image Modeling with Denoising Contrast

[D] Self-Supervised Pretraining Improves Self-Supervised Pretraining

**Summary Of The Paper:**

The paper aims to learn better visual representation by leveraging both contrastive learning (CL) and masked image modeling (MIM). The authors claim that simultaneously performing CL and MIM pertaining will lead to worse results. To mitigate this issue, the paper proposes Layer Grafted Pre-training, which finetunes a pretrained MIM model (e.g., MAE) using contrastive learning (e.g., MoCo-v3 and VICReg). The pretrained models are evaluated on ImageNet with linear and fine-tuning protocols.

**Summary Of The Review:**

Overall, I think the paper is well-written and motivated. But it has limited technical contribution and provides few new insights and knowledge. The method is also not that effective. I tend to give a reject.

---

### Official Review · Reviewer_jbuW · 2022-10-24

**Confidence:** 4
**Correctness:** 3
**Technical Novelty And Significance:** 2
**Empirical Novelty And Significance:** 3
**Recommendation:** 6

**Clarity, Quality, Novelty And Reproducibility:**

The paper is easy to read and understand. It clearly identifies a problem and proposes a solution. The solution is validated empirically in multiple scenarios. I think some experiments may be difficult to reproduce because some implementation details are missing. For example, there are a lot of details about the optimizer but not a lot about the data preprocessing/data augmentation. I encourage the authors to release their code or to add the implementation details in an appendix.


**Strength And Weaknesses:**

**Strengths**
- The paper proposes a new approach to combine both Contrastive learning (CL) and Mask Image Modeling (MIM) to take advantage of the best of both worlds.
- Combining CL and MIM losses does not seem straightforward. An analysis in the paper shows naive joint optimization of CL and MIM losses does not work well because there are some conflicts between the losses. Figure 1 shows the cosine similarity between MIM and CL gradients for each layer. For each layer, there are negative values that indicate the MIM and CL are optimized in opposite directions. The paper also gives a reason to justify these conflicts: the MIM loss is designed to reconstruct the input so it should not be invariant to data augmentations, whereas the CL loss is designed to ensure that the model remains invariant to some data augmentations.
- The paper introduces a simple two-step training approach strategy to reduce the conflicts between MIM and CL losses. First, the model is trained with a MIM loss. Second, the model is fine-tuned with a CL loss. The proposed strategy is not specific to a Vision Transformer architecture and can be used with multiple Vision Transformer architectures. An empirical analysis shows that the order is important so the model should be trained with the MIM loss and it is not a good idea to start with the CL loss.
- The proposed approach is evaluated on ImageNet in multiple scenarios: linear, 1% few-shot, 10% few-shot, and fine-tuning. The proposed approach improves performances in these scenarios for both ViT-B/16 and ViT-L/16 architectures.


**Weaknesses**
- The paper does not introduce new string technical contributions. The proposed approach relies on existing techniques, but the combination and context seem novel.
- The proposed approach seems specific to Vision Transformer architectures and may not adapt to other model architectures like ConvNets. *[post-rebuttal] After the author answer, it does not seem to be a weakness because it can be used with ConvNets.*
- I think adding studying the transfer learning performances will improve the quality of the paper. It is quite standard to study how the learned representations transfer to some downstream tasks. I think analysing the transfer learning performances will be more interesting than the visualization for feature space (Figure 3). *[post-rebuttal] The authors added new transfer learning experiments to answer this concern. Table C shows the performances of an UperNet with different pre-training methods.*


Minor: there is an error at the end of the intro: “performing MIM->CL will considerably damage the performance.” It should be “CL->MIM”


**Summary Of The Paper:**

This paper introduces a new approach to combine both Contrastive learning (CL) and Mask Image Modeling (MIM) to take advantage of the best of both worlds. An analysis shows that a simple combination of the losses does not work well because there are some conflicts between the losses. To solve this problem, the paper introduces a two-step training approach strategy where the model is first trained with a MIM loss, and then fine-tuned with a CL loss. The proposed approach is evaluated on ImageNet in multiple scenarios: linear, 1% few-shot, 10% few-shot, and fine-tuning. The proposed approach improves performances in these scenarios.

**Summary Of The Review:**

Overall, I like the idea of combining both Contrastive learning (CL) and Mask Image Modeling (MIM) to take advantage of the best of both worlds. A problem to combine CL and MIM losses is well identified and a solution is proposed, but the proposed solution relies on existing techniques.

------------------------------------------

[initial post rebuttal]

After reading all the reviews, the rebuttal and the updated paper, I decrease my score to 5. Overall, I think the idea of combining MIM and CL is interesting and well motivated in the paper. However, as point out in multiple reviews, the performance gains on some benchmarks do not seem significant. In the fine-tuning setting, which is popular setting, the proposed approach is 0.8pt worst than C-MAE (Table A). The performance gain seems significant only in the few shots and linear settings. I think it is important to see a more consistent and significant gains on most of the benchmarks to validate the proposed approach.

------------------------------------------

I updated my review and rating to ignore the recent works and to take into account the linear and few-shot settings are the main targets of the paper. I would suggest to update the paper to reflect it. For example, the abstract is not clear about this point.

---

### Official Review · Reviewer_iQDY · 2022-11-01

**Confidence:** 4
**Clarity, Quality, Novelty And Reproducibility:** See my comments above.
**Correctness:** 3
**Technical Novelty And Significance:** 2
**Empirical Novelty And Significance:** 2
**Recommendation:** 5

**Strength And Weaknesses:**

Strengths:
1. The proposed method is simple and easy to follow. The writing is clear.
2. Experimental results show improvement over models either trained by MIM or CL separately on the few shot learning setting.

Weaknesses:
1. While the simplicity of the method is desirable, the novelty of the proposed method is relatively limited. It is well known that combining CL and MIM cannot bring significant improvement, as found in [a,b]. In the paper, the authors only compare performance with [a] without a deep analysis of their differences. Reference [b] is missing. The authors are suggested to compare their method in more detail with existing methods to clarify their contributions, e.g. why their strategy is better than other methods.

[a] C. Tao, et al. Siamese image modeling for self-supervised vision representation learning. arXiv preprint arXiv:2206.01204, 2022.

[b] Z. Huang, et al. Contrastive masked autoencoders are stronger vision learners. arXiv preprint arXiv:2207.13532, 2022.

2. The motivation for solving conflict gradients of MIM/CL may be problematic and need further verification. There are several factors ignored in current experiments which make the conclusion less convincing.
- The gradient discrepancy is not severe according to Fig.1: most numbers are still positive, which may imply the gradients may not be the root cause.
- Do those statistical results remain the same across the whole tranining process? Or it is only like this at specific training point (e.g. 100 epoch as in Fig. 1). In other words, MIM and CL may not always have gradient conflict, so the effect of gradient issue may be mistakenly enlarged.

3. The performance is not competitive when compared with existing methods. The finetune result on ImageNet is the same as original MAE, and worse than [a,b] and others like iBot, showing the limits of applying this method in real use. Compared with finetune results, few shot results are less concerned since the paper mainly claims a better pipeline/training strategy over initial MAE and CL.

**Summary Of The Paper:**

In this paper, the authors propose a simple method to combine masked image modeling (MIM) and contrastive learning (CL) into a joint image pretraining framework. Specifically, MIM and CL are applied to the intermediate block and the end block respectively during training. Two steps are conducted: MIM is applied first and then its learning rate is decreased and CL is added into training.


**Summary Of The Review:**

See my comments above.

---

### Official Review · Reviewer_z4dF · 2022-11-26

**Confidence:** 5
**Correctness:** 4
**Technical Novelty And Significance:** 4
**Empirical Novelty And Significance:** 3
**Recommendation:** 8

**Clarity, Quality, Novelty And Reproducibility:**

This paper is well-written with high quality, and novelty.

The whole algorithm is reproducible.

**Strength And Weaknesses:**

The strength of this submission can be summarized as follows

1)	The authors systematically studied the combination of CL and MIM and analyzes the conflict in the gradient direction that incurs the incompatibility of the two paradigms.

2)	Different from the straightforward combination, the authors proposed to reconcile the conflicts of two paradigms via layer grafting, namely, a cascade way. Specifically, the authors show the order the layer grafting make the different effect, where MIM contributes more in the lower layer and CL contributes more in the higher lower. Then, by means of their advantages of CL and MIM, the authors introduce the layer-grafted pretraining method to enhance representation learning.

3)	A range of experiments have demonstrated that the layer-grafted pretraining significantly improves the performance of either CL or MIM, and achieves the new state-of-the-art compared with the current strong baselines including MoCoV3 and SIM. More extensive analysis of LR search or ablation study as well as the finetuning, and segmentation further confirms the designing intuition and advantages of layer-grafted techniques.

4)	The submission is well written with clear logic, and the topic of this work is well motivated based on sufficient evidence on the direct combination.

Minor weakness

1)	LR search directly reflects the grafting degree among layers, which can be studied further in other different datasets to confirm whether it consumes much empirical experience.

2)	The results of the straightforward combination of MIM and CL can be added to tables when compared with the state-of-the-art baselines. This will strengthen the advantages of the proposed layer-grafting techniques.

3)	Layer-grafting pretraining for more general scenarios can be discussed to motivate more explorations in the future.


**Summary Of The Paper:**

This paper studies an interesting layer-grafting technique to promote the combination of CL and MIM for representation learning. Specially, the authors find that directly combining such two paradigms shows the negative influence due to the gradient conflicts, and discovers that the sequential cascade in a certain order can address this problem and consistently improve the performance of representation learning.

**Summary Of The Review:**

The paper provides a new road for representation learning. The observed phenomenon is interesting and important. Then, a novel method is proposed.

---

> ### Author Response · Authors · 2022-11-26
> **Response to reviewer z4dF**
>
> Thank you for your valuable comments!
>
> **Q1**: LR search directly reflects the grafting degree among layers, which can be studied further in other different datasets to confirm whether it consumes much empirical experience.
>
> **A1**: That’s a great suggestion. We are running the experiments and would incorporate them into the future version.
>
> **Q2**: The results of the straightforward combination of MIM and CL can be added to tables when compared with the state-of-the-art baselines. This will strengthen the advantages of the proposed layer-grafting techniques.
>
> **A2**: Thanks for your kind reminder. We would update the draft accordingly.
>
> **Q3**: Layer-grafting pretraining for more general scenarios can be discussed to motivate more explorations in the future.
>
> **A3**: Great suggestion! Here we discuss two general scenarios for the proposed method and potential future directions:
>
> * Explore this finding on language tasks: similar to self-supervised learning in vision tasks, CL [a] and Mask Language Modeling (MLM) [b] also demonstrate different properties in language tasks. While MLM provides a strong init point, CL further yields stronger zero or few-shot learning performance. It would be interesting to explore whether the layer-wise complementary effect also holds for CL and MLM in the language tasks.
>
> * The few-shot transferability: while current CL methods excel on in-domain few-shot learning, it suffers from a large performance drop when transferring to a different domain [c]. By grafting CL on the top of MIM, the proposed method not only ensures strong few-shot in-domain performance but also inherits the strong transferability of the MIM method. Therefore, we conjecture that the proposed method may also empower the learned representation with few-shot transferability improvement. One interesting future direction is to verify this hypothesis.
>
> [a] Rethmeier, Nils, and Isabelle Augenstein. "A Primer on Contrastive Pretraining in Language Processing: Methods, Lessons Learned & Perspectives." ACM Computing Surveys (CSUR) (2021).
>
> [b] Devlin, Jacob, et al. "Bert: Pre-training of deep bidirectional transformers for language understanding." arXiv preprint arXiv:1810.04805 (2018).
>
> [c] Li, Suichan, et al. "Improve Unsupervised Pretraining for Few-label Transfer." Proceedings of the IEEE/CVF International Conference on Computer Vision. 2021.

---

> > ### Comment · Reviewer_z4dF · 2022-11-27
> > **Thanks for addressing my concers**
> >
> > Thanks for addressing my concerns. However, after reading your response, I have the following questions.
> >
> > 1. CL is not very common in NLP tasks. It is worth discussing the complementary effects between CL and MLM in the real-world language problems.
> >
> > 2. As for the few-shot transferability, if MIM has a strong ability to transfer knowledge, why do we need to combine CL and MIM? More discussions should be added. Besides, in the image processing field, CL can indeed enhance the few-shot transferability. The statement in the part of few-shot transferability might not be true. What is the meaning of "in-domain" in your response?

---

> > > ### Author Response · Authors · 2022-11-28
> > > **Following discussion**
> > >
> > > Thanks for the timely reply and inspiring questions. Here are our responses.
> > >
> > > **Q1**: CL is not very common in NLP tasks. It is worth discussing the complementary effects between CL and MLM in real-world language problems.
> > >
> > > **A1**: Take the sentence pair regression tasks like clustering or semantic search as an example, while MLM can learn the token-wise context feature, it fails to learn the semantically meaningful sentence embeddings, which leads to an inferior pairing performance compared to supervised learning ones. In contrast, CL enables the learning of global sentence embeddings via enforcing the alignment among sentence pairs [a]. Moreover, MLM can also complement CL by providing a strong understanding of the context.
> > >
> > > We conjecture that the layer-wise complementary effect may still hold for the sentence pair regression tasks given that MLM and CL here play a similar role with MIM and CL in vision, respectively: i) Both MLM and MIM capture the token-wise local features ii) CL in both vision and language learns the semantically-aware grouping and invariance.
> > >
> > > **Q2-1**: Why we need CL:
> > >
> > > **A2-1**: The stronger transferability of MIM over CL is verified on downstream tasks with adequate data [b]. However, given that MIM fails to achieve a strong performance even for in-domain few-shot learning (refer to Q2-3 for the definition), transfer few-shot performance is very likely to further decrease. Therefore, we want to complement MIM with the strong in-domain few-shot learning of CL. Meanwhile, MIM can benefit CL with stronger transferability. Together, they may achieve an improvement in the few-shot transferability.
> > >
> > > **Q2-2**: CL indeed enhance the few-shot transferability in image processing. The statement in the part about few-shot transferability might not be true
> > >
> > > **A2-2**: Thanks for pointing out this. To clarify, we are talking about transfer few-shot learning of classification tasks, where the performance of the self-supervised pre-training is shown to be much worse than its supervised counterpart [c].
> > >
> > > It is interesting to see that few-shot transfer learning works well in image processing. The reason for the difference may lie in the difference in the domain gap. For instance, in image processing, [d] successfully decreases the domain gap by collecting multilevel noisy images. However, the domain gap of the classification tasks is harder to address and thusly more severe, which may lead to an inferior performance for CL.
> > >
> > > **Q2-3**: The meaning of “in-domain”.
> > >
> > > **A2-3**: Here we use “in-domain few-shot learning” to describe the few-shot task that employs the same data distribution for both pre-training and few-shot fine-tuning. Meanwhile, the pre-training dataset is large-scale and unlabeled while the few-shot fine-tuning dataset is small-scale and labeled. In-domain few-shot learning is also known as semi-supervised learning [e].
> > >
> > >
> > > [a] Zhang, Yan, et al. "An unsupervised sentence embedding method by mutual information maximization." arXiv preprint arXiv:2009.12061 (2020).
> > >
> > > [b] He, Kaiming, et al. "Masked autoencoders are scalable vision learners." Proceedings of the IEEE/CVF Conference on Computer Vision and Pattern Recognition. 2022.
> > >
> > > [c] Li, Suichan, et al. "Improve Unsupervised Pretraining for Few-label Transfer." Proceedings of the IEEE/CVF International Conference on Computer Vision. 2021.
> > >
> > > [d] Jiang, Bo, et al. "Multilevel Noise Contrastive Network for Few-Shot Image Denoising." IEEE Transactions on Instrumentation and Measurement 71 (2022): 1-13.
> > >
> > > [e] Chen, Ting, et al. "Big self-supervised models are strong semi-supervised learners." Advances in neural information processing systems 33 (2020): 22243-22255.

---

> > > > ### Comment · Reviewer_z4dF · 2022-11-29
> > > > **Thanks for the reply**
> > > >
> > > > I would like to express my thanks to the authors.
> > > >
> > > > My concerns are fully addressed now.

---

### Author Response · Authors · 2022-11-29
**Summary of review & response**

We thank the efforts of all reviewers during the rebuttal and discussion period. Here, we provide a summary.

We start by summarizing the strengths of the proposed work recognized by reviewers:

* Reviewers z4df, jbuW, and nkwf all think positively of our novelty by mentioning that “A novel method is proposed”, “empirical finding is valuable to be shared” and “combination and context seem novel”.

* Reviewers z4df, iQDY, jbuW, and nkwf affirm the significant improvement of the proposed method by mentioning that “the layer-grafted pretraining significantly improves the performance of either CL or MIM, and achieves the new state-of-the-art compared with the current strong baselines including MoCoV3 and SIM”, “Experimental results show improvement over models either trained by MIM or CL separately on the few-shot learning setting.”, “The proposed approach is evaluated on ImageNet in multiple scenarios: linear, 1% few-shot, 10% few-shot, and fine-tuning … for both ViT-B/16 and ViT-L/16 architectures.”, and “The proposed method outperforms the state-of-the-art self-supervised learning baselines.”

* Reviewers z4df and jbuW commend the way that we analyze the conflict of MIM and CL by mentioning “The authors systematically studied the combination of CL and MIM and analyze the conflict in the gradient direction that incurs the incompatibility of the two paradigms.” and “An analysis in the paper shows naive joint optimization of CL and MIM losses does not work well because there are some conflicts between the losses.”

* Last but not least, Reviewers z4df, iQDY, Yr9u, and nkwf all endorse that this submission is well-written and easy to follow.

The main point that sparked reviewer critiques and discussions was:

* The reviewers considered that our significant performance improvement only occurs for few-shot evaluation, but the gains are smaller on “full-data” fine-tuning

We have clarified the point by elaborating:

* Few-shot learning is our main focus, and an important long-standing target that has been actively tackled by existing self-supervised learning literature [a,b,c,d].

* Few-shot learning is also fine-tuning (a type of it): just fine-tuning using fewer labels. It is also unnecessary to contrast “fine-tuning” and “few-shot learning” as they are actually one: few-shot learning is a type of fine-tuning, just using fewer labels. We probably shouldn’t understand fine-tuning narrowly as using all labels of the downstream datasets. Instead, tuning with fewer labels is more practically useful. Our performance gains on the main few-shot setting are always significant.

* Even on full-data fine-tuning, our results still surpass MIM, CL, and other strong baselines with consistent advantages (except one concurrent arxiv work, which we have pointed out in a previous response by citing reviewer guidelines)

After discussion, we are happy to see that several reviewers now agree “the linear and few-shot settings are the main targets of the paper” and consider their concerns as addressed.

Lastly, we want to thank AC and the reviewers again for their active engagement: the discussions have definitely helped improve our draft a lot. We will continue standing by and actively addressing new questions if there are any.

[a] Chen, Ting, et al. "Big self-supervised models are strong semi-supervised learners." Advances in neural information processing systems 33 (2020): 22243-22255.

[b] Zhai, Xiaohua, et al. "S4l: Self-supervised semi-supervised learning." Proceedings of the IEEE/CVF International Conference on Computer Vision. 2019.

[c] Su, Jong-Chyi, Subhransu Maji, and Bharath Hariharan. "When does self-supervision improve few-shot learning?." European conference on computer vision. Springer, Cham, 2020.

[d] Ericsson, Linus, Henry Gouk, and Timothy M. Hospedales. "How well do self-supervised models transfer?." Proceedings of the IEEE/CVF Conference on Computer Vision and Pattern Recognition. 2021.

---

### Decision · Program_Chairs · 2023-01-20

**Decision:**

Accept: poster

**Justification For Why Not Higher Score:**

According to my expertise and reviewing process, this paper should belong to an Accept with poster.

**Justification For Why Not Lower Score:**

According to my expertise and reviewing process, this paper should belong to an Accept with poster.

**Metareview: Summary, Strengths And Weaknesses:**

This paper studies an interesting layer-grafting technique to promote the combination of CL and MIM for representation learning. Specially, the authors find that directly combining such two paradigms shows the negative influence due to the gradient conflicts, and discovers that the sequential cascade in a certain order can address this problem and consistently improve the performance of representation learning. A range of experiments have demonstrated that the layer-grafted pretraining significantly improves the performance of either CL or MIM, and achieves the new state-of-the-art compared with the current strong baselines including MoCoV3 and SIM. More extensive analysis of LR search or ablation study as well as the finetuning, and segmentation further confirms the designing intuition and advantages of layer-grafted techniques. For example, this paper has shown its strengths on some benchmarks, including linear evaluation, few-shot classification, and ADE20K segmentation.

The clarity and novelty are obviously above the bar of ICLR. While the reviewers had some concerns on the limitation of performance, the authors did a particularly good job in their rebuttal. Note that there are two points to be merged into the updated version: 1) if the few-shot setting is main focus, this paper needs to present more analysis on why the method is specifically useful for the few-shot and the differences in features required between few-shot and fine-tuning; 2) if the linear and few-shot settings are the main targets of the paper, the authors are strongly suggested to update the paper to reflect it. For example, the abstract and title are not clear about this point, and should be updated. Overall, after a zoom meeting, all of us have made a consensus to accept this paper for publication! Please include the additional experimental results and further explanation in the next version.

**Note From Pc:**

if the above contains the word "oral" or "spotlight" please see: "oral" presentation means -> notable-top-5% and "spotlight" means -> notable-top-25%. As stated in our emails, we are disassociating presentation type from AC recommendations

**Summary Of Ac-Reviewer Meeting:**

After discussing with Reviewers z4dF, nkwf, iQDY and Yr9u via zoom, our feeling is quite positive to the current version and rebuttal. The clarity and novelty are obviously above the bar of ICLR. Since this is a top-tier machine learning conference paper, the most important thing is to judge whether this paper is novel or not. Although the proposed method is marginally better than some methods based on some benchmarks, its merits are shown on the other benchmarks as well. Thus, in general, this paper contributes a useful and interesting method to the field.

By discussion, there are two points to be merged into the updated version: 1) if the few-shot setting is main focus, this paper needs to present more analysis on why the method is specifically useful for the few-shot and the differences in features required between few-shot and fine-tuning; 2) if the linear and few-shot settings are the main targets of the paper, the authors are strongly suggested to update the paper to reflect it. For example, the abstract and title are not clear about this point, and should be updated.

Overall, after a zoom meeting, all of us have made a consensus to accept this paper for publication!